# Multi-Level Fusion Temporal–Spatial Co-Attention for Video-Based Person Re-Identification

**DOI:** 10.3390/e23121686

**Published:** 2021-12-15

**Authors:** Shengyu Pei, Xiaoping Fan

**Affiliations:** 1School of Automation, Central South University, Changsha 410075, China; shengyupei@csu.edu.cn; 2School of Information Technology and Management, Hunan University of Finance and Economics, Changsha 410205, China

**Keywords:** video-based person re-identification, multi-level fusion, temporal–spatial co-attention, knowledge evolution

## Abstract

A convolutional neural network can easily fall into local minima for insufficient data, and the needed training is unstable. Many current methods are used to solve these problems by adding pedestrian attributes, pedestrian postures, and other auxiliary information, but they require additional collection, which is time-consuming and laborious. Every video sequence frame has a different degree of similarity. In this paper, multi-level fusion temporal–spatial co-attention is adopted to improve person re-identification (reID). For a small dataset, the improved network can better prevent over-fitting and reduce the dataset limit. Specifically, the concept of knowledge evolution is introduced into video-based person re-identification to improve the backbone residual neural network (ResNet). The global branch, local branch, and attention branch are used in parallel for feature extraction. Three high-level features are embedded in the metric learning network to improve the network’s generalization ability and the accuracy of video-based person re-identification. Simulation experiments are implemented on small datasets PRID2011 and iLIDS-VID, and the improved network can better prevent over-fitting. Experiments are also implemented on MARS and DukeMTMC-VideoReID, and the proposed method can be used to extract more feature information and improve the network’s generalization ability. The results show that our method achieves better performance. The model achieves 90.15% Rank1 and 81.91% mAP on MARS.

## 1. Introduction

Person re-identification (reID) uses computer vision to determine whether a particular pedestrian is present in an image or video sequence. Person re-identification is widely considered a sub-problem of image retrieval. Given a monitored pedestrian image, we retrieve the pedestrian’s photo under the cross-camera. Person re-identification can combine with pedestrian detection and pedestrian tracking technology to be applied to state-of-the-art video surveillance, advanced security, and other fields. Person re-identification can be divided into image-based and video-based person re-identification. In recent years, many significant achievements have been made in image-based person re-identification. Video data contain more information than image data. Recently, video-based person re-identification has attracted more attention. Video-based person re-identification aims to match personal video with candidate galleries. Given a pedestrian video, the purpose of video-based person re-identification is to identify the same pedestrian from videos captured by different cameras. How to embed the video’s temporal and spatial information into the feature representation is a challenge. Presently, most methods cannot be used to ascertain the relationship between frames in feature extraction fully.

In recent years, scholars have proposed many different methods to solve video-based person re-identification [1]. The authors of [2] use the advantage of the affinity between all available query images and gallery images to present a novel group-shuffling random walk. The authors of [3,4] integrate attitudes to extract the representations of different viewpoints. The authors of [5] prove that the common practice of aggregating temporal and spatial features by recurrent neural network (RNN) structures may not be optimal. The authors of [6] use the relationship between different parts to propose an adaptive structure-aware adjacency graph. The authors of [7] improve generator and constrained two-level fusion networks through generative adversarial networks (GANs). The authors of [8] present a 3D-guided adversarial transform (3D-GAT) network to explore the migration ability of source training data.

The current state-of-the-art video person re-recognition is based on deep convolutional networks. With robust deep convolutional networks and large-scale labeling benchmarks, these methods achieve good performance and efficiency. Although significant progress has been made in video person re-identification, most current algorithms do not fully use the rich temporal and spatial information in the video. Due to imperfect person detection algorithms, some generated bounding boxes are smaller or larger than the actual situation. Figure 1a shows images generated by a person detection algorithm, and it may cause some misalignment problems. Because the sequence frames of pedestrians in a video are highly similar, current methods perform the same operation on each frame, so these methods usually produce highly redundant features for the video sequence frames. Figure 1b shows an activation map generated by ordinary video person re-recognition algorithms. The backpack brings in the most excellent attention, but it is not easy to distinguish between two pedestrians. Therefore, it is necessary to explore a method that can fully mine the temporal–spatial information of the video and discover different situations in different frames to form the complete characteristics of each identity.

In this article, we propose a multi-level fusion temporal–spatial co-attention network to explore the temporal–spatial information of video sequence frames fully. Figure 1d shows that MLTS contains the global module, local module, and attention module. The steps are as follows: Extract the overall features of the identity in the video through the global module. Correct the misalignment of the bounding box through the local module. The attention module can extract the complementary features of consecutive frames in the video. Figure 1c is the activation map generated by the algorithm in this paper. Solve the problem of misalignment of the bounding boxes of the two frames of images. The local module divides the bounding box equally and makes judgments, solving the problem of bounding box misalignment. The attention module pays attention to the time information between these frames, explores the different features of consecutive frames, and finds the complementary characteristics of successive frames.

A video sequence can be used as a feature vector, but the degree of similarity for each video sequence frame is different. Each video sequence has additional semantic information in temporal and spatial space. Inspired by the above idea, we propose a new multi-level fusion temporal–spatial co-attention (MLTS). The contributions of this paper are as follows:This paper proposes a multi-level fusion temporal–spatial co-attention strategy to solve the video-based person re-identification problems according to the different similarity degrees of video-sequence frames.This paper introduces the knowledge evolution technique to improve the feature extraction performance of the backbone. Experiments show that this strategy is effective for a video-based person re-identification dataset.The network has excellent performance on four video person re-identification datasets. Compared with the latest results, the network proposed in this paper is optimal in some metrics.

The following is the contents of the rest of this paper. Section 2 introduces related work of video-based person re-identification. Section 3 provides our new proposed method. Section 4 provides the proposed method’s simulation experiments on four video-based person re-recognition datasets. Section 5 provides a summary of our new proposed method.

## 2. Related Work

### 2.1. Video-Based Person Re-Identification

Video-based person re-identification has attracted attention [9]. Scholars try to introduce pedestrian attributes, pedestrian posture, and other auxiliary information to help solve the video-based person re-identification problems [10,11,12,13]. However, these methods require additional information, usually through the different collections, which is time-consuming and laborious. There are also some methods that convert video sequences into image pairs, and use image-based person re-recognition models [14,15,16,17] to mine video sequence frame features. There are also some 2D or 3D convolution methods based on recurrent neural networks (RNNs) [18] to deal with the problem of feature extraction of video sequence frames. These methods show considerable effectiveness but ignore the temporal information of the video data. Moreover, these methods lack sufficient generalization ability for small datasets. In this paper, we try to develop an improved model to extract more effective temporal–spatial feature representations to solve the problem of video-based person re-identification. Our algorithm also has good performance on small datasets PRID2011 and iLIDS-VID.

### 2.2. Based Spatial–Temporal

For video-based person re-recognition, the video contains multiple image sequence frames. Multiple image sequence frames have both temporal and spatial relationships [19,20,21]. Wang et al. [22] proposed a Spatial–Temporal Aggregation Module (STAM) to generate the discriminative features. It contains a spatial reference attention and a temporal reference attention. Aich et al. [23] proposed a Spatial–Temporal Representation Factorization (STRF) to learn complementary spatial–temporal feature representations. Liu et al. [24] proposed a Spatial–Temporal Correlation and Topology Learning (CTL) framework to learn discriminative representation. STRF and CTL are based on 3D convolutional networks. These network models lack the learning of local features of related frames. The effect is evident in the case of slow changes between frames, but deviation will occur in the case of rapid changes between frames. However, although it is rich in temporal and spatial information, information also increases the difficulty of solving the problem. For example, video is much noisier than a static image and is easily affected by occlusion, changes in the scene, and overlapping pedestrians. If the video is processed frame by frame, the computer hardware is very demanding, and the calculation cost is very high. Video features contain time sequence and spatial information, so they are sensitive to classifiers and prone to over-fitting. However, these methods perform the same operation on each frame, resulting in a high degree of redundancy in the features of different frames. This paper mainly improves the spatial–temporal aggregation mechanism of video sequence frames to avoid the loss of pedestrian features in some frames.

## 3. Proposed Method

To solve the problem of video-based person re-identification, we propose a multi-level fusion network structure. Our network framework is shown in Figure 2. In particular, (1) we use ResNet50 based on knowledge evolution as the backbone to extract features, (2) we first use the global branch network to extract the global-level feature embedding, then use the local branch network to extract the part-level feature embedding, and then use the self-attention branch network to obtain the relationship between the local and the global, and extract the temporal–spatial feature embedding, and (3) we use a unified learning method to calculate the loss to optimize the model and update the feature embedding until the optimal performance is obtained.

### 3.1. ResNet50 with Knowledge Evolution

The effect of neural network training depends on a large dataset. With a small dataset, the network is prone to fall into local extreme values. This paper introduces the knowledge evolution (KE) [25] method, which can improve the performance and reduce the inference cost of learning to the network. KE reduces overfitting and reduces the burden of data. To reduce the reference cost, [25] proposed a customized splitting technology for a CNN to minimize the reference cost, namely kernel-level convolutional-aware splitting (KELS).

For each layer *l*, the binary mask $M_{l}$ divides *l* into two specific parts: The fit hypothesis and the reset hypothesis. Given a resolution rate $s_{r}$ and a convolution filter $F_{l}\in R^{{\hat{C}}_{o}\times \kappa \times \kappa \times C_{i}}$, a fit hypothesis is introduced to include the first $\left\lceil s_{r}\times C_{i}\right\rceil$ kernel in the first $\left\lceil s_{r}\times C_{o}\right\rceil$ filter. KELS guarantees dimensional matching between the resulting convolution filters. This paper integrates KELS in the classic network architecture ResNet.

### 3.2. Multi-Level Fusion Temporal–Spatial Attention

Figure 2 shows the network framework proposed in this paper. “A” represents the global branch. “B” represents the local branch. “C” represents the attention branch. They are shown in Figure 3, Figure 4 and Figure 5. The detail is shown in Algorithm 1.

We use three different branch strategies in feature fusion: Figure 3 represents the global network with spatial attention, Figure 4 illustrates the local network, and Figure 5 represents the temporal–spatial co-attention network.

In this section, we introduce the proposed network framework in detail. The network framework consists of three branches: The global branch, the local branch, and the attention branch, which correspond to “A”, “B”, and “C”, respectively, in Figure 2. The video itself consists of different frames. One frame represents an image. The input video is an image sequence with a fixed number of frames. In the training phase, the input images sequence is preprocessed, which plays the role of data enhancement. The data enhancement used in this paper includes random horizontal flip, padding, random crop, normalization, random erasing, etc. Then, data are fed into the KE ResNet backbone network architecture used in this paper. An introduction to KE is described in Section 3.1. In Layer 4 of KE ResNet, we copy two copies: One into the global and the local branch network, the other into the attention branch network.
**Algorithm 1** MLTS algorithm.**Input:** Epoch *epochs*;  1:Video sequence *D*;  2:Learning rate $lr=3.5\times 10^{-5}$;  3:**for** each i∈[1,epochs] **do**  4:    Extract feature vectors from input videos according to Equations (Equation 4) and (Equation 6);  5:    Predict labels from input videos by the model;  6:    **if** i≤10 **then**  7:        $lr=lr\times \frac{i}{10}$  8:    **else if** 10<i≤40 **then**  9:        $lr=3.5\times 10^{-4}$10:    **else if** 40<i≤70 **then**11:        $lr=3.5\times 10^{-5}$12:    **else if** 70<i≤200 **then**13:        $lr=3.5\times 10^{-6}$14:    **end if**15:    Update loss with Equation (Equation 8)16:    **if** $Rank_{i}>Rank_{best}$ **then**17:        $Rank_{best}=Rank_{i}$,18:        Update best model;19:    **end if**20:**end for**
**Output:** Best model, mAP, cmc.


#### 3.2.1. Global Branch

The global branch network is shown in Figure 3. The image sequences are fed into an adaptive three-dimensional average pooling layer. First, we need to exchange the time-series dimension of feature embedding with the channel dimension to obtain the global feature embedding of the current input image’s all-time series. The input contains the frame image sequence; high-level image features are aggregated, and the spatial features can be extracted through the module.

#### 3.2.2. Local Branch

The local branch network is shown in Figure 4. The feature embedding fed by the local branch network is consistent with that of the global branch. Image feature segmentation is implemented in the spatial–temporal dimension because the video sequence strongly correlates with spatial–temporal information. It helps to maintain the relevance between the pedestrian image frames.

#### 3.2.3. Attention Branch

The attention branching network is shown in Figure 5. The attention branching network is also divided into two sub-branches. One of the sub-branches is a two-dimensional mean-pooling layer, which obtains the timing feature and embeds it into $\left\{\phi_{j}^{i}\right\}$, i∈[1,t]. Its size is t×d, where *t* represents the time sequence length, and *d* represents the image’s horizontal feature dimension. In this paper, the attention mechanism is used to aggregate the temporal features, as shown in Equation (Equation 1). It is another sub-branch “D” as shown in Figure 5. The main function of the sub-branch network is to obtain the temporal attention score. A scalar vector $s_{j}^{i}$ is obtained through two-dimensional convolution and one-dimensional convolution operations. It represents the score of the *i*th frame of the *j*th sequence. From this scalar vector, we can finally calculate the temporal attention score as shown in Equation (Equation 2) or Equation (Equation 3).
(1)$$\phi_{j}=\frac{1}{t}\sum_{i=1}^{t}\alpha_{j}^{i}\phi_{j}^{i}$$
(2)$$\alpha_{j}^{i}=\frac{e^{s_{j}^{i}}}{\sum_{k=1}^{t}e^{s_{j}^{k}}}$$
(3)$$\alpha_{j}^{i}=\frac{\sigma (s_{j}^{i})}{\sum_{k=1}^{t}\sigma (s_{j}^{k})}$$

Here, the Softmax function is adopted in Equation (Equation 2) obtain the time sequence attention score. Equation (Equation 3) uses the sigmoid function to obtain the time sequence attention score. σ(·) is the sigmoid function. Finally, according to Equation (Equation 1), the results of these two sub-branches will be used to obtain the final feature embedding. In the evaluation phase, these three branches of A, B, and C are normalized by L1, concatenated together, and finally added into Equations (Equation 4) and (Equation 6).

### 3.3. Unified Learning Mechanism

In the video-based person re-identification task, we can use $D=\{\left(x_{1},y_{1}\right),(x_{2}$$,y_{2}),\ldots ,$$\left(x_{n},y_{n})\right\}$ to represent the sets of the identity label, where $x_{i}=\{x_{i}^{1},x_{i}^{2},\ldots ,x_{i}^{t}\}$ is the *i*th input video sequence, and *t* is the length of the original input video sequence. $x_{i}^{j}$ represents the *j*th frame image of the *i*th input video sequence. The length of the input video sequence is set as t=8, that is, 8 frames of images are randomly selected from the video sequence. Therefore, the first *i*th input video sequence can be set as $x_{i}=\left\{x_{i}^{1},x_{i}^{2},x_{i}^{3},x_{i}^{4},x_{i}^{5},x_{i}^{6},x_{i}^{7},x_{i}^{8}\right\}$.

#### 3.3.1. Identity Learning

For a set *D* of the identity label, we can define target functions based on the identity label. The first is a classification function based on the identity label.
(4)$$F_{Id}=\min_{w_{Id},\theta }\sum_{i=1}^{n}L_{Id}\left(f_{Id}\left(w_{Id},\phi \left(\theta ,x_{i}\right)\right),y_{i}\right)$$

Here, ϕ represents the feature embedding function, $\phi (\theta ,x_{i})$ represents the image feature embedding of the first *i*th identity, and θ represents the training parameters of the feature embedding function. $f_{Id}\left(w_{Id},\phi \right)$ represents the classification function of the feature embedded in the identity label, and $w_{Id}$ represents the training parameters of the classification function. $L_{Id}\left(z_{i},y_{i}\right)$ represents the image label loss function of the *i*th identity. The purpose of the function $F_{Id}$ is to find the appropriate image feature embedding and make the identity obtained by training as consistent as possible with the ground truth label.
(5)$$L_{Id}=-\sum_{i=1}^{n}y_{i}\log q_{i}$$

Here, *q* represents the confidence probability of the predicted label, $q_{i}=\frac{e^{z_{i}}}{\sum_{j=1}^{n}e^{z_{j}}}$, *z* is the predicted labels of model. *y* is the ground truth labels. *n* is the batch size.

#### 3.3.2. Metric Learning

For the video-based reID, we use metric learning for feature embedding to narrow the distance between the positive samples and push the distance between the negative samples.
(6)$$F_{Tri}=\min_{\theta }\sum_{i=1}^{n}L_{Tri}\left(\phi \left(\theta ,x_{i}\right),y_{i}\right)$$

$L_{Tri}\left(\phi ,y_{i}\right)$ represents the metric loss function of the feature embedded in the identity label. The purpose of the function $F_{Tri}$ is to find the appropriate image feature embedding so that the identities of the same labels are as close as possible and the identities of different labels are as separate as possible.
(7)$$L_{Tri}=log\left(1+e^{d(an,po)-d(an,ne)}\right)$$

We use soft metric learning. Here, *d* is the distance function. For each anchor an, we can find the hardest positive po and negative ne.

We combine classification learning with metric learning to find the appropriate image feature embedding to better solve video-based person re-identification problems.

In the evaluation phase, given a video sequence containing the same pedestrian from the query set, the video-based person re-identification task is to find the video sequence of the same pedestrian in all video sequences from the gallery set. Each video sequence from the query set can be defined as: $Q=\{q_{1},q_{2},\ldots ,q_{t_{q}}\}$. $t_{q}$ is the number of frames of the original video sequence, and $q_{i}$ is the first *i*th image of the video sequence. $G=\{g_{1},g_{2},\ldots ,g_{m}\}$ represents the set of all video sequences from the gallery set, and *m* is the total number of video sequences from the gallery set. The first *j*th video sequence from the gallery set can be defined as: $g_{j}=\left\{g_{j}^{1},g_{j}^{2},\ldots ,g_{j}^{t_{g}}\right\}$. $t_{g}$ represents the number of frames of the original video sequence of the first *j*th sequence. The video sequence contains 8 frames of images, so the length of the video sequence is designed as: $t_{q}=t_{g}=8$.

#### 3.3.3. Loss Function

We use unified learning to optimize the model. The loss function is defined as follows:(8)$$L=L_{Id}\left(G\right)+\sum_{i=1}^{6}L_{Id}\left(L_{i}\right)+L_{Id}\left(A\right)+L_{Tri}\left(G\right)+\sum_{i=1}^{6}L_{Tri}\left(L_{i}\right)+L_{Tri}\left(A\right)$$

Here, *G* represents the global branch output, *L* represents the local branch output, and *A* represents the attention branch output. In the local branch, we calculated six different component losses.

Under the unified learning mechanism, we calculate the sum of global branch loss, local branch loss, and attention branch loss and update network parameters through backpropagation.

## 4. Experiment

### 4.1. Dataset and Setting

We evaluate the effectiveness of the proposed methods in the widely used actual person re-identification task with real-world pedestrian video datasets. DukeMTMC-VideoReID, MARS have more videos and are more challenging.

MARSThe MARS dataset (motion analysis and person re-identification) is currently the largest video-based person re-identification dataset [26]. It is an extended version of the Market-1501 dataset. Because all bounding boxes and tracks are automatically generated, it contains distractions, and each identity may have multiple tracks.DukeMTMC-VideoReIDThe DukeMTMC-VideoReID dataset [27] is a subset of the Duke Multi-Target, Multi-Camera (DukeMTMC) tracking dataset used for video-based person re-identification. This dataset contains 702 identity training sets, 702 identity test sets, and 408 identity interference items. The training set has 2196 videos, and the test set has a capacity of 2636 videos. Each video contains images of pedestrians captured in every 12 frames. During testing, one video from each identity is used for query, and the rest of the videos are treated as a gallery.PRID2011The Austrian Institute of Technology created the Person Re-ID 2011 (PRID2011) dataset [28] to promote the development of gender recognition for pedestrians. This dataset consists of images extracted from multiple pedestrian tracks recorded by two different static surveillance cameras. It has 385 videos from camera A and 749 from camera B, with only 200 identities showing up in both cameras.iLIDS-VIDThe iLIDS-VID dataset [29,30,31,32] has 600 videos from the the Imagery Library for Intelligent Detection Systems (i-LIDS) Multiple-Camera Tracking Scenario (iLIDS-MCTS) dataset containing 300 identities. It is based on the assumption that every identity in a natural person re-identification system should have a track. It is taken from the monitoring air reception hall and creates this dataset from two disjointed cameras containing static images (iLIDS-VID) and image sequences (iLIDS-VIS). Due to the limitation of the iLIDS-MCTS dataset, the occlusion of the iLIDS-VID dataset is very critical.

For the fairness of comparison, the backbone is ResNet50 in this paper. Each video sequence’s frame image has a width of 128 pixels and a height of 256 pixels. Unless otherwise specified, the length of the video sequence is 4.

### 4.2. Evaluation Metrics

As for the evaluation metrics, this paper adopts several standard metrics most commonly used in person re-identification. Cumulative matching curve (CMC), mean average precision (MAP), and receiver operating characteristic (ROC) curve are included.

### 4.3. Comparison with the State of the Art

To verify the effectiveness of the proposed method, MLTS, we execute many experiments in this section.

Table 1, Table 2 and Table 3 show the performance comparison of the current video-based person re-identification methods in the four datasets. Baseline is the benchmark method mentioned in this article. MLTS refers to a multi-level network with three different branches added. It can be seen from the three tables that the baseline performance is limited. MLTS with three different branches has been added, and the performance has improved significantly. The multi-level network can learn more about spatial–temporal feature representation. Table 1 lists the comparison of evaluation metrics of the current mainstream methods for the MARS dataset. Our proposed method in this paper is superior to the method based on mAP, Rank-1, Rank-5, and Rank-20. The latest Rank-1 score of this paper is 90.15%. Table 2 shows the performance comparison of the proposed method with other methods on the DukeMTMC-VideoReID dataset. From Table 2, we can see that our proposed method also has advantages in the evaluation metrics mAP, Rank-1, Rank-5, and Rank-20. Table 3 also shows the performance comparison between the proposed method and the latest methods on the PRID2011 and iLIDS-VID dataset. These two datasets are small, and the network easily falls into local extreme values. From Table 3, we can see that our proposed method is superior to other methods for the iLIDS-VID dataset. The latest Rank-1 score of this paper is 94.0%. For the PRID2011 dataset, our proposed method also has advantages in the evaluation metrics Rank-1 and Rank-5. The latest Rank-1 score of this paper is 96.63%. The experimental results also verify that the algorithm in this paper does not have a strong dependence on the data. It can also prevent over-fitting and avoid restrictions that are only effective for big datasets.

### 4.4. Time Complexity Analysis

By analyzing the proposed network structure, the time complexity will be given in this section. It is shown in Equations (Equation 9)–(Equation 12).
(9)$$T_{g}=T\left[A_{g}\left(L4\right)\right]+T\left[C\left(E_{g}\right)\right]+T\left[L\left(E_{g}\right)\right]$$
(10)$$T_{l}=T\left[A_{l}\left(L4\right)\right]+T\left[C\left(E_{l}\right)\right]+T\left[L\left(E_{l}\right)\right]$$
(11)$$T_{a}=T\left[A_{a}\left(L4\right)\right]+T\left[C\left(E_{a}\right)\right]+T\left[L\left(E_{a}\right)\right]$$
(12)$$T_{MLTS}=T_{g}+6\times T_{l}+T_{a}$$
where *T* is the time function, $T_{g}$ is the time complexity of the global branch, $T_{l}$ is the time complexity of the local branch, and $T_{a}$ is for the attention branch. *A* is the aggregation function, *C* is the classification function, and *L* is the loss function. $A_{g}$ represents the aggregation function of the global strategy; $A_{a}$ is for the attention strategy. L4 represents the output of layer 4 from ResNet50. $E_{g}$, $E_{l}$, and $E_{a}$ represent global branch feature embedding, local branch feature embedding, and attention branch feature embedding, respectively. Because the pedestrian image is cut laterally into 6 equal parts in this paper, $T_{MLTS}$ is the sum of $T_{g}$, $T_{a}$ and six times $T_{a}$.

### 4.5. Ablation Study

In Table 4, our ablation experiments on MARS and DukeMTMC-VideoReID dataset are listed. Three different strategies were used in our ablation experiments. They are global branch strategy only (“G”), local branch strategy only (“L”), attention branch strategy only (“A”), and the combination of these strategies. There are three evaluation metrics, namely mAP, Rank-1, and Rank-5. It can be seen from the table that both the MARS dataset and DukeMTMC-VideoReID dataset use a three-branch co-evaluation strategy to achieve better results than other strategies.

In Table 5, our ablation experiments on the PRID2011 and iLIDS-VID dataset are listed. Our ablation experiments also used three different branches. “G” represents the global branch strategy, “L” represents the local branch strategy, and “A” represents the attention branch strategy. There are three evaluation metrics, namely Rank-1, Rank-5, and Rank-20. As can be seen from the table, whether it is the PRID2011 dataset or the iLIDS-VID dataset, the performance of the three strategies’ combination is significantly improved compared with either branch strategy. For the PRID2011 dataset, using the combination of three strategies improved the Rank-1 metric by 5.62%. For the iLIDS-VID dataset, our proposed method improved its Rank-1 by 5.33%. This also proves the effectiveness of our proposed method.

There are three different strategies in Figure 6. “G+L+A” represents the baseline, namely the global branch, local branch, and attention branch, as proposed in this paper. “G+L+A+8” means that the video sequence’s length is changed from 4 to 8 on the baseline. “G+L+A+KE+8” represents the introduction of the knowledge evolution method into ResNet50, and the video sequence length is 8. The blue bar is the Rank-1 metric; the orange bar is the mAP metric. Using the knowledge evolution method and a considerable video sequence length for the MARS dataset can improve Rank-1 and MAP metrics by 0.11% and 1.25%, respectively. For the DukeMTMC-VideoReID dataset, after using knowledge evolution and a considerable video sequence length, the two evaluation metrics are significantly improved. The Rank-1 metrics can increase by 1.14%, and the mAP metrics can increase by 1.71%. It is shown that selecting the appropriate video sequence length and knowledge evolution method will be helpful in the performance improvement.

Figure 7 shows the four datasets’ ROC curves. In the figure, the blue line represents the ROC curve for the “G+L+A” strategy. The yellow line represents our presented method. For the DukeMTMC-VideoReID and MARS dataset, we found that the proposed algorithm is better than the “G+L+A” strategy.

For the PRID2011 and iLIDS-VID datasets, we used the same comparison. As can be seen from the figure, in the top half of the rank, our proposed method performs better than the “G+L+A” strategy but slightly behind the “G+L+A” strategy in the middle of the rank. The final rank of these methods is 100%.

Figure 8 shows the experimental results of four strategies on four datasets. The numbers from 0 to 3 represent policies “A”, “G”, “G+A”, and “G+ L+A”, respectively. Figure 8a shows the influence of these four different strategies on dataset DukemtMC-VideoReID and MARS. Figure 8b shows the effects of these four different strategies on datasets iLIDS-VID and PRID2011. For each dataset, strategy “G” performed better than “A” on the mAP curve, and strategy “G+A” performed better than “A” and “G” alone. “G+L+A” is the best.

### 4.6. Visualization

To better express the proposed method’s effectiveness, some visualization results are presented in this paper. They are presented in Figure 9 and Figure 10. Figure 9 shows the focus area’s visualization of a pedestrian video in the DukeMTMC-VideoReID and MARS datasets by our proposed network. We visualized each dataset using two different strategies: (a) represents the heat map results using the “G” strategy. The first line represents the original four-frame continuous images of the pedestrian video, and the second line represents the visualization results with “G” strategy; (b) represents the heat map results using the “G+L+A” strategy. The first line is the original pedestrian video, and the second line represents the visualization results with the “G+L+A” strategy.

Figure 10 shows the heat map results of the PRID2011 and iLIDS-VID dataset by the method in this paper: (a) represents the heat map results using the “G” strategy; (b) represents the heat map results using the “G+L+A” strategy. Because the algorithm randomly and continuously takes four frames of images from the pedestrian video, the original video of (a) and (b) is slightly different. This does not prevent a visual comparison of the two different strategies. From the heat image, we can see where the network is focused. As can be seen from Figure 9 and Figure 10, no matter whether using DukeMTMC-VideoReID, MARS, iLIDS-VID, or PRID2011, the algorithm in this paper has a more broad and abstract focus area for pedestrian images, which can better express the semantic information of images.

## 5. Conclusions

In this paper, we propose a multi-level fusion temporal–spatial co-attention person re-identification. Our method demonstrates the reliable performance on four large-scale datasets. The introduction of knowledge evolution’s ResNet backbone can improve feature extraction performance. The designed local branch, global branch, and attention branch network can pay more attention to the information of video frames. Of course, there is still a gap between this and actual use. We hope that our method can provide an option for video-based person re-identification applications.

The multi-level fusion mechanism and unified learning that we propose are general and are not limited to specific areas of pedestrian re-identification. Future aims are as follows:In our method, we use the original unified learning mechanism. In the future, we will continue to explore the influence and further optimize this unified learning mechanism.In ResNet, we use mean aggregate features. In the future, we will continue to explore the influence of the mean aggregation method.Our proposed method is not only for video re-identification tasks but also for other supervision tasks. These all need to be explored.

## Figures and Tables

**Figure 1 entropy-23-01686-f001:**
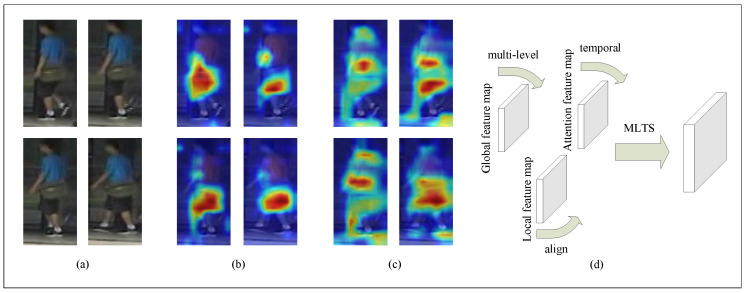
An example of activating maps of video sequences. The darker the color, the higher the value. (**a**) Video sequence frames, (**b**) Activation map of ordinary methods, (**c**) Activation map of our method, (**d**) MLTS structure.

**Figure 2 entropy-23-01686-f002:**
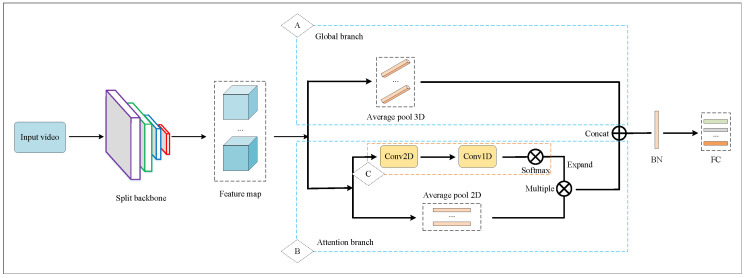
Network structure.

**Figure 3 entropy-23-01686-f003:**
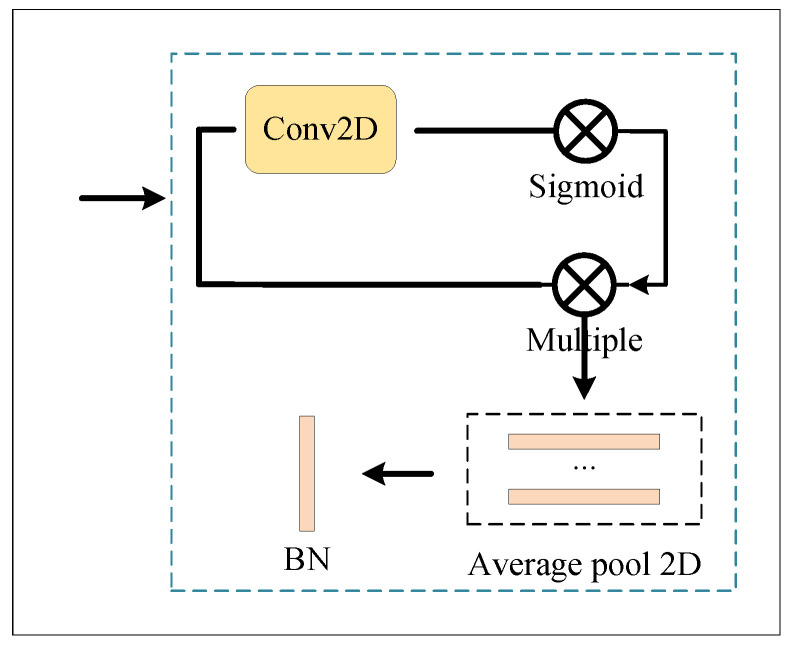
Global module.

**Figure 4 entropy-23-01686-f004:**
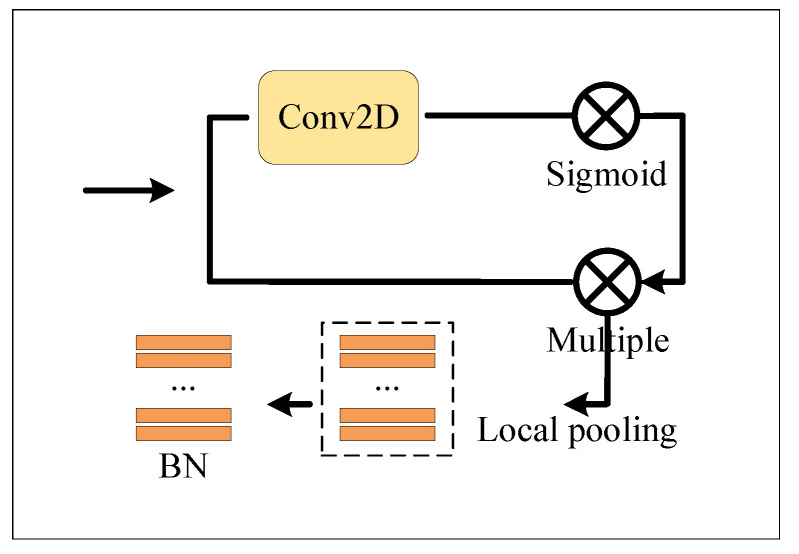
Local module.

**Figure 5 entropy-23-01686-f005:**
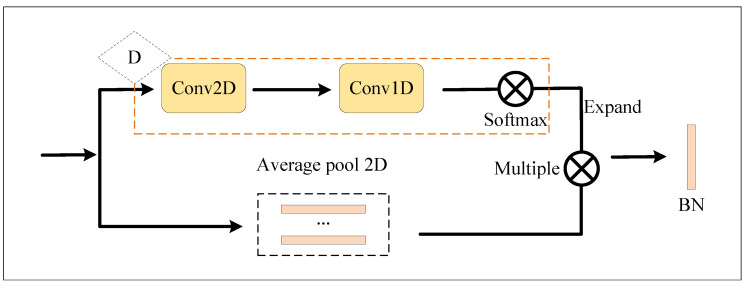
Attention module.

**Figure 6 entropy-23-01686-f006:**
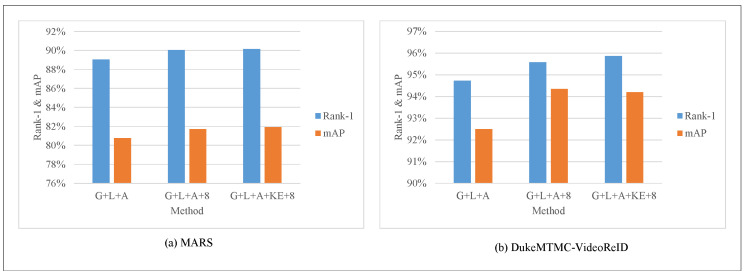
Ablation study on MARS and DukeMTMC-VideoReID.

**Figure 7 entropy-23-01686-f007:**
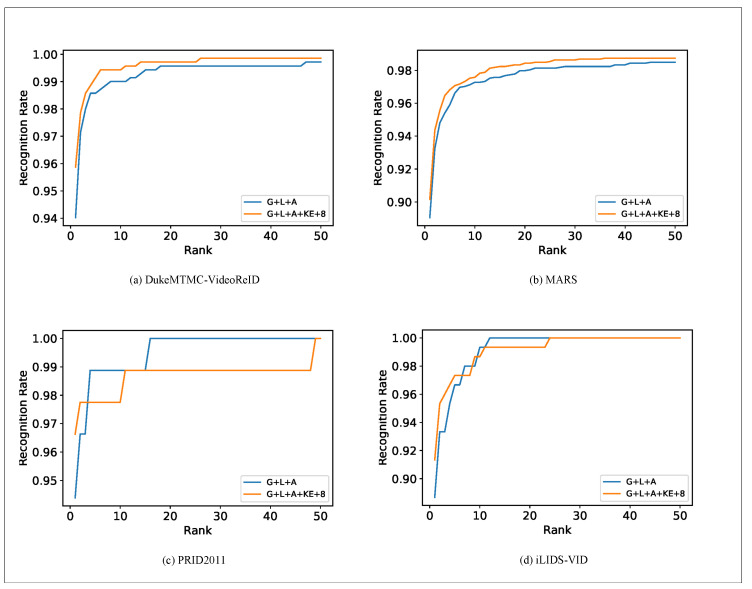
ROC curve on DukeMTMC-VideoReID, MARS, PRID2011, and iLIDS-VID.

**Figure 8 entropy-23-01686-f008:**
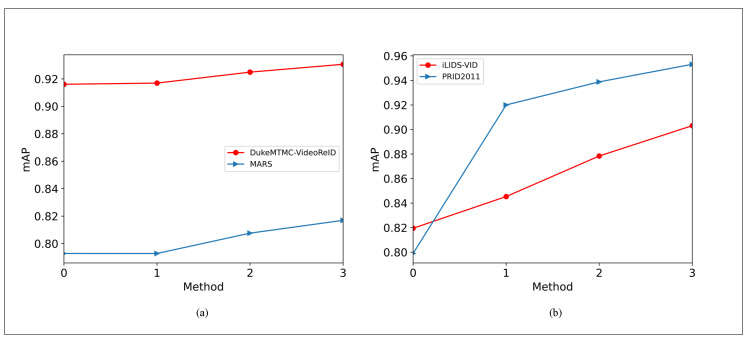
mAP curve on DukeMTMC-VideoReID, MARS, PRID2011, and iLIDS-VID.

**Figure 9 entropy-23-01686-f009:**
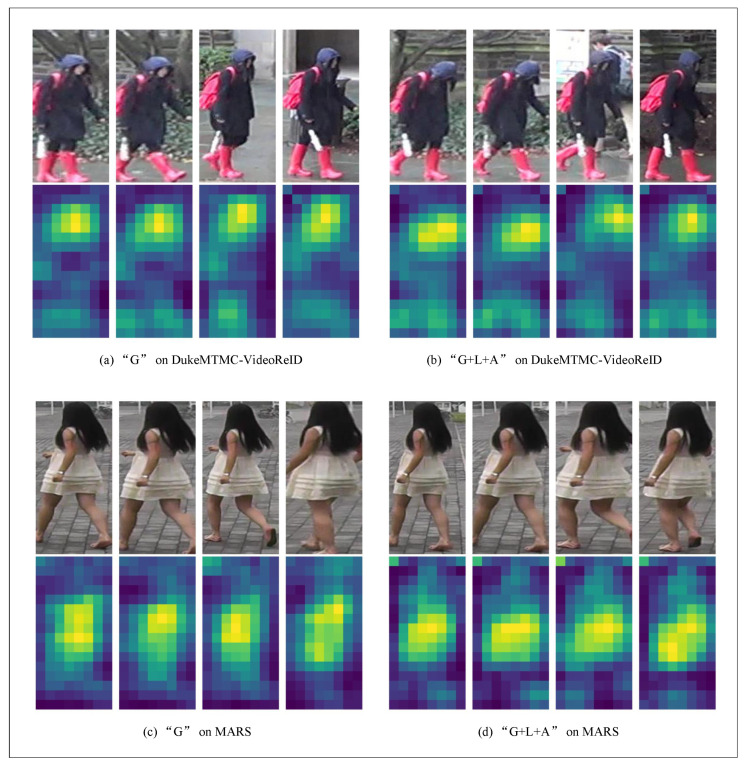
Visualization on DukeMTMC-VideoReID, MARS.

**Figure 10 entropy-23-01686-f010:**
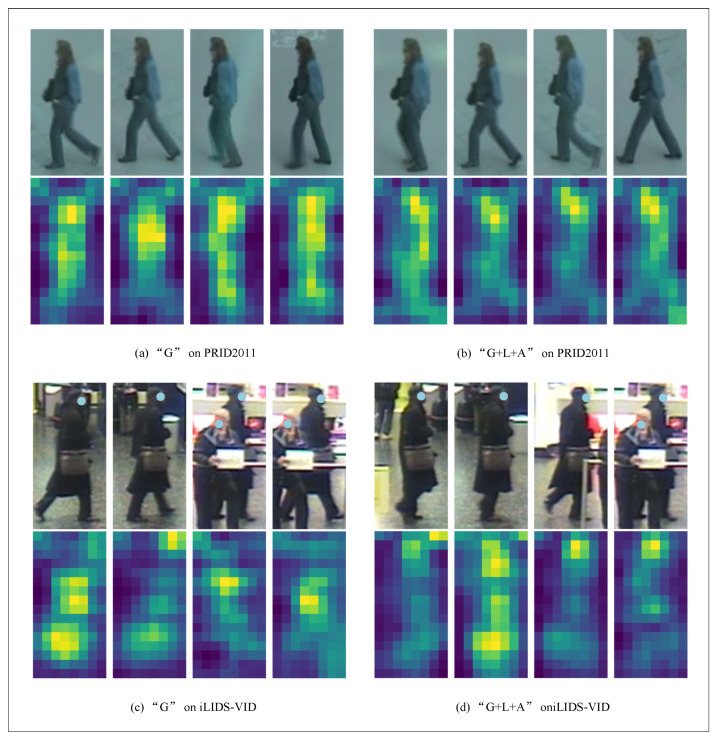
Visualization on PRID2011 and iLIDS-VID.

**Table 1 entropy-23-01686-t001:** Comparison with the state-of-the-art algorithms on MARS.

Method	mAP	Rank-1	Rank-5	Rank-20
BoW+kissme [26]	15.50	30.60	46.20	59.20
IDE+XQDA [26]	47.60	65.30	82.00	89.00
SeeForest [33]	50.70	70.60	90.00	97.60
QAN [34]	51.70	73.70	84.90	91.60
DCF [35]	56.05	71.77	86.57	93.08
TriNet [36]	67.70	79.80	91.36	-
MCA [37]	71.17	77.17	-	-
DRSA [38]	65.80	82.30	-	-
DuATM [39]	67.73	81.16	92.47	-
MGCAM [37]	71.17	77.17	-	-
PBR [40]	75.90	84.70	92.80	95.00
CSA [41]	76.10	86.30	94.70	98.20
STMP [42]	72.70	84.40	93.20	96.30
M3D [43]	74.06	84.39	93.84	97.74
STA [44]	80.80	86.30	95.70	98.10
GLTR [45]	78.47	87.02	95.76	98.23
FT-WFT [46]	82.9	88.6	-	-
STGCN [47]	**83.7**	89.9	-	-
DPRM [48]	83.0	89.0	96.6	98.3
MLTS (Baseline)	79.28	87.93	95.45	97.76
MLTS	81.91	**90.15**	**96.82**	**98.43**

**Table 2 entropy-23-01686-t002:** Comparison with the state-of-the-art algorithms on DukeMTMC-VideoReID.

Method	mAP	Rank-1	Rank-5	Rank-20
ETAP-Net [27]	78.34	83.62	94.59	97.58
STA [44]	**94.90**	**96.20**	99.30	99.60
GLTR [45]	93.74	96.29	**99.30**	99.71
MLTS (Baseline)	91.69	94.02	98.58	99.05
MLTS	94.20	95.87	99.15	**99.72**

**Table 3 entropy-23-01686-t003:** Comparison with the state-of-the-art algorithms on PRID2011 and iLIDS-VID.

Method	PRID2011		iLIDS-VID	
	Rank-1	Rank-5	Rank-1	Rank-5
BoW+XQDA [26]	31.80	58.50	14.00	32.20
IDE+XQDA [26]	77.30	93.50	53.00	81.40
DFCP [49]	51.60	83.10	34.30	63.30
AMOC [12]	83.70	98.30	68.70	94.30
QAN [34]	90.30	98.20	68.00	86.80
DRSA [38]	93.20	-	80.20	-
RCN [50]	70.00	90.00	58.00	84.00
DRCN [51]	69.00	88.40	46.10	76.80
RFA-Net [52]	58.20	85.80	49.30	76.80
SeeForest [33]	79.40	94.40	55.20	86.50
T-CN [53]	81.10	85.00	60.60	83.80
CSA [41]	93.00	99.30	85.40	96.70
STMP [42]	92.70	98.80	84.30	96.80
M3D [43]	94.40	100.00	74.00	94.33
GLTR [45]	95.50	**100.00**	86.00	98.00
TCLNet [54]	-	-	84.3	-
STRF [23]	-	-	89.3	-
CTL [24]	-	-	89.7	-
Xuehu Liu [55]	96.2	99.7	90.4	98.3
W3AN [56]	95.8	99.5	89.2	98.1
MLTS (Baseline)	91.01	95.51	82.00	93.33
MLTS	**96.63**	97.75	**94.0**	**98.67**

**Table 4 entropy-23-01686-t004:** Ablation study on MARS and DukeMTMC-VideoReID.

Components			MARS			DukeMTMC-VideoReID		
G	L	A	mAP	Rank-1	Rank-5	mAP	Rank-1	Rank-5
		🗸	79.28	87.88	96.01	91.61	93.73	98.86
🗸			79.28	87.93	95.45	91.69	94.02	98.58
🗸		🗸	80.76	89.04	95.91	92.49	94.73	99.00
🗸	🗸	🗸	81.91	90.15	96.82	94.20	95.87	99.15

**Table 5 entropy-23-01686-t005:** Ablation study on PRID2011 and iLIDS-VID.

Components			PRID2011			iLIDS-VID		
G	L	A	Rank-1	Rank-5	Rank-20	Rank1	Rank-5	Rank-20
		🗸	77.53	88.76	95.51	79.33	93.33	96.67
🗸			91.01	95.51	98.88	82.00	93.33	99.33
🗸		🗸	94.38	98.88	100.00	88.67	96.67	100.00
🗸	🗸	🗸	96.63	97.85	98.88	94.0	98.67	100.00

## Data Availability

The four datasets used in this study can be found here: https://github.com/liangzheng06/MARS-evaluation, https://github.com/Yu-Wu/DukeMTMC-VideoReID, https://www.tugraz.at/institute/icg/research/team-bischof/lrs/downloads/prid11/, https://xiatian-zhu.github.io/downloads_qmul_iLIDS-VID_ReID_dataset.html (accessed on: 19 October 2021).

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
