# Peer review of "Multi-Level Fusion Temporal–Spatial Co-Attention for Video-Based Person Re-Identification"

_entropy, 2021, doi:10.3390/e23121686_

Round 1

Reviewer 1 Report

Line 17: clearly mention the quantitative performance gains.

Line 258: Need a paragraph justifying why these particular datasets were selected. Also need to discuss the unique characteristics and challenges in each dataset with respect to the problem. So that readers understand how the new model works better in diverse settings.

Line 212: provide detail network architecture.

Line 298: the name of the method is defined in page 8. this name should be introduced much earlier in the paper.

Table 3: Why Rank 20 and mAP results are missing in this table? In Table 1, 2, 3 highlight best performances (bold) for each performance measure.

Line 390: Conclusion sounds like introduction. This needs complete rewrite highlighting key outcomes and potential applications and benefits. Future work is too generic, suggest some clear direction you are considering.

Reviewer 2 Report

The paper presents a multi-level fusion temporal-spatial co-attention strategy to solve the video-based person re-identification problems. The author basically fuses a few network components to get the feature embedding.

Weakness of the paper:

  •  The paper lack novelty as it uses existing techniques to achieve a multi-level fusion strategy.
  • The paper does not present the usual pipeline of writing a paper. For instance:
    • There is no problem definition in the introduction section.
    • The related work section does not provide a comparative analysis of the state-of-the-art works. In addition to that, it does not provide any explanation regarding how the proposed approach differs from the existing state-of-the-art approaches.
    • In the proposed method section, it states that it introduces the knowledge evolution (KE) [19] method, but there is no dedicated section to explain it.
    • The experimental analysis does not consider any baseline over which the proposed method is developed.

The authors should address all the above-mentioned issues.

Round 2

Reviewer 1 Report

Well done with the revised version.

Author Response

Agreed. Thanks for your comments. We have revised the manuscript and corrected related expressions and grammatical errors.

Reviewer 2 Report

Although the paper is modified with the given direction, still there are some major things that are needed to be considered. For example:

  • There is no problem definition in the introduction section. In the abstract section, the paper mentioned about preventing over-fitting and reducing the dataset limit. But in the introduction and the consecutive section, they do not mention anything regarding how their method prevents over-fitting and reduces the dataset limit. The authors are suggested to follow the style of following paper where the suggested paper is dealing with misalignment problem: https://arxiv.org/pdf/2007.08434.pdf

  • Related works section needed to be updated based on "how the proposed MLTS method differs from existing works?".
  • Thanks for improving the experimental section.

Round 3

Reviewer 2 Report

I am satisfied with the current version of the paper. Please check the spelling and consistency before moving to the final version of the paper